# Structural variation at the maize *WUSCHEL1* locus alters stem cell organization in inflorescences

Zongliang Chen [1], Wei Li[1], Craig Gaines[2], Amy Buck[1,2], Mary Galli [1] & Andrea Gallavotti [1,3] ✉

Structural variation in plant genomes is a significant driver of phenotypic variability in traits important for the domestication and productivity of crop species. Among these are traits that depend on functional meristems, populations of stem cells maintained by the CLAVATA-WUSCHEL (CLV-WUS) negative feedback-loop that controls the expression of the WUS homeobox transcription factor. WUS function and impact on maize development and yield remain largely unexplored. Here we show that the maize dominant *Barren inflorescence3* (*Bif3*) mutant harbors a tandem duplicated copy of the *ZmWUS1* gene, *ZmWUS1-B*, whose novel promoter enhances transcription in a ring-like pattern. Overexpression of *ZmWUS1-B* is due to multimerized binding sites for type-B RESPONSE REGULATORs (RRs), key transcription factors in cytokinin signaling. Hypersensitivity to cytokinin causes stem cell overproliferation and major rearrangements of *Bif3* inflorescence meristems, leading to the formation of ball-shaped ears and severely affecting productivity. These findings establish *ZmWUS1* as an essential meristem size regulator in maize and highlight the striking effect of cis-regulatory variation on a key developmental program.

[1] Waksman Institute of Microbiology, Rutgers University, Piscataway, NJ 08854-8020, USA. [2] Section of Cell and Developmental Biology, University of California San Diego, La Jolla, CA 92093-0116, USA. [3] Department of Plant Biology, Rutgers University, New Brunswick, NJ 08901, USA. ✉email: agallavotti@waksman.rutgers.edu

The control of plant stem cells is essential for sustaining the function of apical meristems, plant growth, and ultimately productivity[1]. Stem cells reside at the growing tip of meristems, where they differentiate to produce new organs throughout the life of plant and maintain a constant reservoir of pluripotent stem cells. Stem cell maintenance in the shoot is under the control of the CLAVATA-WUSCHEL (CLV-WUS) negative feedback-loop, which is tightly integrated with hormone function, in particular auxin and cytokinin that promote cell differentiation and proliferation, respectively[2]. WUS is a homeodomain transcription factor (TF) produced in the organizing center (OC) domain of apical meristems and is transported via plasmodesmata into the apical domain (called central zone, CZ) to promote proliferation of stem cells[3]. Flanking the CZ is the peripheral zone (PZ) where cells are recruited for the initiation of new lateral primordia. In the CZ, WUS activates the CLV3 gene, encoding a short signaling peptide perceived by a series of leucine-rich repeat (LRR)-receptor-like complexes, among which are complexes containing CLV1 and CLV2. Activation of CLV3 in the OC is prevented by the action of WUS in conjunction with the GRAS-transcription regulators HAIRY MERISTEMs (HAMs)[4,5]. A distinct ZmFCP1-FEA3 ligand–receptor combination, originally identified in maize, prevents WUS gene expression in the region below the OC (also called rib zone, RZ), thus confining WUS expression within the OC of meristems[6].

The molecular architecture of the CLV-WUS pathway is conserved across different plant species[7], but individual gene contributions differ and, in some cases, genes acquire distinct functions[8,9]. In particular, based on studies in rice and Arabidopsis, it is believed that WUS function is not evolutionary conserved between eudicot and monocot species[9–11], but this hypothesis is solely based on work carried out in rice. The function of the two uncharacterized maize co-orthologs of WUS, ZmWUS1 and ZmWUS2, is unknown[8]. Recently, interest in ZmWUS's stem cell promoting properties has resurfaced due to their use in efficient transformation systems for maize and other recalcitrant plant species[12,13].

The regulation of WUS transcription is crucial for meristem homeostasis, whereby high WUS expression leads to enlarged meristems and low expression leads to the formation of small meristems[3,14]. The phytohormone cytokinin induces WUS expression, and it was recently shown that type-B Arabidopsis response regulators (B-ARRs) directly bind to the WUS proximal promoter region and activate its expression in a cytokinin-dependent fashion[15–20].

In this study, by characterization of the dominant maize mutant Barren inflorescence3 (Bif3) we identified a tandem-duplicated copy of ZmWUS1 whose expression is dramatically enhanced by the insertion of a short stretch of chimeric proximal promoter sequence. We showed that this causes striking changes in inflorescence meristem (IM) architecture and leads to extreme stem cell proliferation and the formation of spherical meristems, whereby OC, CZ, and PZ domains become arranged in concentric circles. We determined that multimerized binding sites of type-B RR TFs at the duplicated ZmWUS1 locus are the underlying cause of its ectopic expression and promote cytokinin hypersensitivity. These findings refine the gene-regulatory network of maize meristems placing ZmWUS1 at the center of vegetative and reproductive meristem size regulation, thus enhancing our ability to exploit variation in the CLV-WUS pathway for diverse agricultural traits and for fast and efficient transformation systems.

## Results

### Bif3 inflorescences show defects in meristem initiation and maintenance.
The Barren inflorescence3 (Bif3) mutant was originally isolated in an uncharacterized genetic background. To better evaluate its phenotype, we introgressed the only Bif3 allele available (Bif3-ref) in two different maize inbred lines, A619 and B73. In both genetic backgrounds, the Bif3 mutation severely affected inflorescence architecture, but more so in the A619 background where the mutation showed dominance in ear and semi-dominance in tassel phenotypes. In B73, instead, the mutant phenotype appeared slightly milder, in particular in heterozygous tassels, which showed no obvious change in the number of branches and spikelets, structures that contain florets (Supplementary Fig. 1). However, these plants were male sterile, as were homozygous mutants. The male sterility was not due to a gametophytic effect given that the mutation segregated as expected ($X^2$, $0.75 < p > 0.5$, $n = 82$) and pollen germination rate was unchanged (Supplementary Fig. 1), but very few pollen grains were produced.

Given that the phenotype was more severe in A619, we carefully analyzed the developmental defects of Bif3 inflorescences in this genetic background. The central spike of heterozygous and homozygous Bif3 tassels was significantly shorter than wild-type (~67% of wild-type length, Student's t-test $p < 0.0001$, $n \geq 13$), and had several visible barren patches devoid of spikelets (Fig. 1a). Ears, on the other hand, were extremely reduced in size, ball-shaped, and carried only few kernels upon fertilization (Fig. 1b). Scanning electron microscopy (SEM) analysis of 2–3 mm Bif3 tassel and ear primordia revealed the formation of extremely enlarged IMs, spikelet pair meristems (SPMs) developing on the top of the inflorescence, and single rather than paired spikelet meristems (SMs), with bigger than normal subtending suppressed bracts (Fig. 1c, d). In normal inflorescences, IMs form organized rows of axillary meristems (SPMs and paired SMs) in both inflorescences (Fig. 1c, d). While other mutants with increased IM size (fasciated) produce more axillary meristems (AMs)[6,21–26], Bif3 mutants were unique in initiating only a small number of AMs.

To investigate whether other meristems were affected, we also measured vegetative shoot apical meristems (SAMs) in heterozygous and homozygous Bif3 mutants and determined that they were approximately 8% larger than those of wild-type siblings (mean diameter $119.7 \pm 7.7$ μm and $118.9 \pm 5.9$ μm, respectively, vs. $108.7 \pm 6.0$ μm) (Fig. 1e, f). In Bif3 mutants, larger SAMs correlated with leaves that were 20% wider and 12% shorter than wild-type (Supplementary Fig. 1). Overall, these results indicate that Bif3 has defects in vegetative and reproductive meristem size regulation as well as reproductive AM initiation.

### Bif3 is caused by a tandem-duplicated ZmWUS1 gene expressed by a novel chimeric promoter.
To identify the causal gene, we first mapped the Bif3 locus to a small region of ~90 kb on the top of chromosome 2 that contained six genes including ZmWUS1, one of the maize co-orthologs of the Arabidopsis WUS gene (AtWUS)[8] (Fig. 2a). ZmWUS1 encodes a protein of 320 amino acids with 64% identity with AtWUS. By performing RNA-seq analysis of Bif3/Bif3 and wild-type ear primordia (3–4 mm), we subsequently determined that of the six genes within this mapping window, two genes, ZmWUS1 (GRMZM2G047448) and its immediate downstream gene GRMZM2G047321, were differentially expressed in Bif3/Bif3 (16.5-fold and 1.6-fold higher relative to wild-type, respectively; Fig. 2b). Given the enlarged IM phenotype, we concentrated on ZmWUS1 and confirmed its strong up-regulation by qRT-PCR (Fig. 2c). These results suggested that the enhanced expression of ZmWUS1 may cause the Bif3 dominant mutant phenotype.

To identify the molecular basis of elevated ZmWUS1 expression levels, we conducted a series of PCR and Southern blot

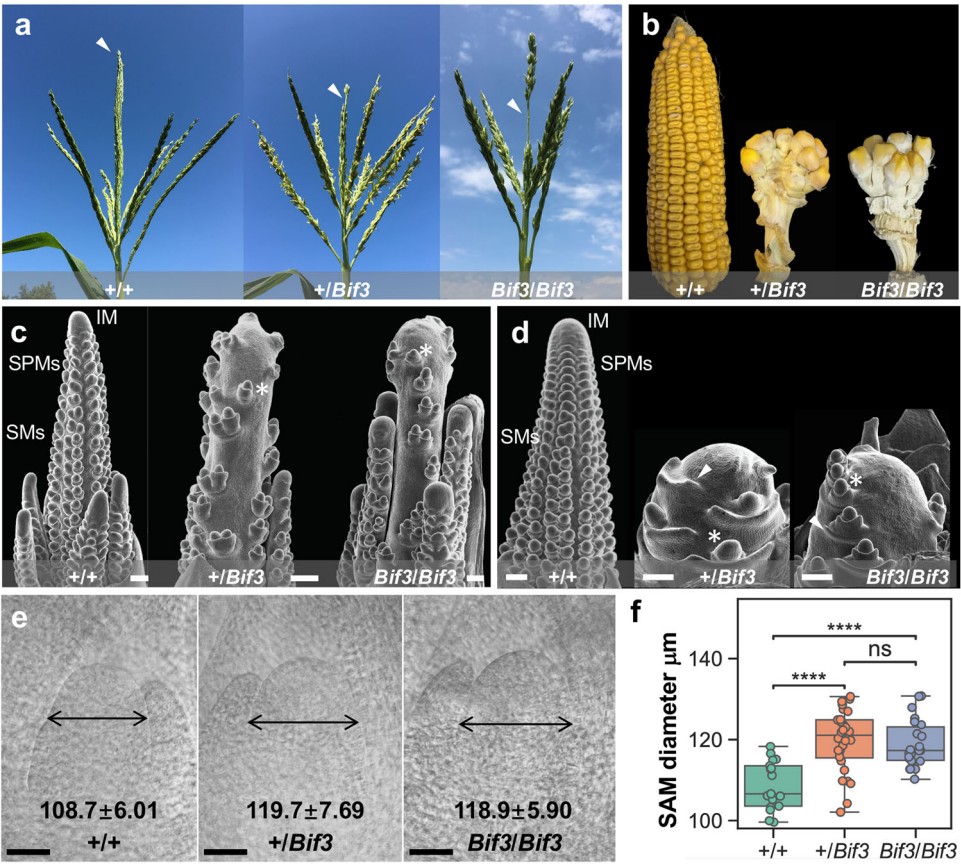

**Fig. 1 Bif3 mutant phenotypes. a** Normal tassels produce spikelets on branches and central spikes (white arrowheads). Barren patches are observed in *Bif3* mutants. **b** Ball-shaped ear of *Bif3* mutants. **c, d** SEM images of wild-type and *Bif3* tassel, and ear primordia. *Bif3* mutants show enlarged IMs, a reduced number of SPMs, single SMs (asterisks) and enlarged suppressed bracts (arrowheads). **e** Cleared shoot apical meristems (SAMs) from wild-type (A619), heterozygous, and homozygous *Bif3* plants. The *Bif3* SAMs have larger diameter (double-headed arrows). **f** Quantification of SAM diameter: two-tailed Student's *t*-test, ****$p < 0.0001$ ($n_{+/+} = 17$; $n_{+/Bif3} = 28$; $n_{Bif3/Bif3} = 22$). Box plot center line, median; box limits, upper and lower quartiles; whiskers, maximum and minimum values. Scale bars: 200 μm in (**c** and **d**); 50 μm in (**e**).

analyses that revealed an additional copy of *ZmWUS1* (hereafter referred to as *ZmWUS1-B* to distinguish it from the original copy, *ZmWUS1-A*) at the *Bif3* locus (Fig. 2d and Supplementary Fig. 2). Using a PCR-based chromosome walking method[27], we sequenced both flanking regions of *ZmWUS1-B* and detected a tandem duplication of a 16 kb fragment containing *ZmWUS1-B*, the two downstream genes (*GRMZM2G047321* and *GRMZM2G047018*), and a fragment of the third downstream gene *GRMZM2G046968* (including promoter, first exon, and partial first intron) (Fig. 2d). The tandem-duplicated *ZmWUS1* locus also contained copies of a *MuDR* transposable element, the autonomous element of the *Mutator* transposon family[28]. The 5′ junction of the tandem-duplicated regions contained a novel 119 bp region. While the coding sequences of both copies of the *ZmWUS1* gene were identical, the upstream regulatory region of *ZmWUS1-B* included the original 444 bp proximal promoter region of *ZmWUS1*, the 119 bp junction sequence, and a partial fragment of the *GRMZM2G046968* gene (Fig. 2d). The *ZmWUS1-B* gene was only present in the *Bif3* mutant background, and not in a diverse panel of inbred lines (Supplementary Fig. 2). By sequencing several DNA sequence stretches in the region, we determined that Ki3, a Thai inbred, was very similar, though not identical (~99.1% identity; Supplementary Fig. 3), to the original *Bif3* mutant background, suggesting that the *Bif3* mutation arose in a closely related tropical line.

To test whether the overexpression of *ZmWUS1* was the cause of the *Bif3* mutant phenotype, we performed a targeted ethyl-methane

sulfonate (EMS) intragenic suppressor screen in the A619 background, to ask if knock-out mutations in the *ZmWUS1* locus could restore a wild-type phenotype (reversion). Pollen from homozygous *Bif3* mutants was treated with EMS and used to fertilize ears of wild-type A619. Approximately 10,000 M1 plants were screened and we identified 512 phenotypically wild-type plants (revertants) that were heterozygous for *Bif3*. The unexpected high frequency of reversion (~100-fold higher than expected[29]) led us to investigate its nature. We first asked if any of the M1 revertant individuals carried an EMS-induced mutation in *ZmWUS1-A* or *ZmWUS1-B*, using a 3D-pooling strategy and next-generation sequencing. This approach identified one revertant individual (*Bif3-Rev1*) in which the *ZmWUS1-B* gene carried a G > A point mutation in the start codon (Fig. 3a and Supplementary Fig. 2). As a consequence of this mutation, the predicted ZmWUS1-B protein lost the N-terminus, including a section of the DNA-binding domain. In other revertants, we first performed PCR on the original pools to determine whether the *ZmWUS1-B* copy was still present. A small subset of revertants was then characterized by a combination of PCRs and Southern blots. This approach led to the identification of another revertant (*Bif3-Rev2*) which still carried the *ZmWUS1-B* gene. However, in this plant *ZmWUS1-B* contained a longer native promoter (1178 bp proximal promoter instead of 444 bp proximal promoter of *ZmWUS1-B*) and lost the 119 bp junction region (Fig. 3a and Supplementary Fig. 2). A subset of other revertants (*Bif3-Rev3* to *Rev20*) were tested with a series of markers designed to distinguish the duplicated regions, and all had lost the *ZmWUS1-B* duplicated

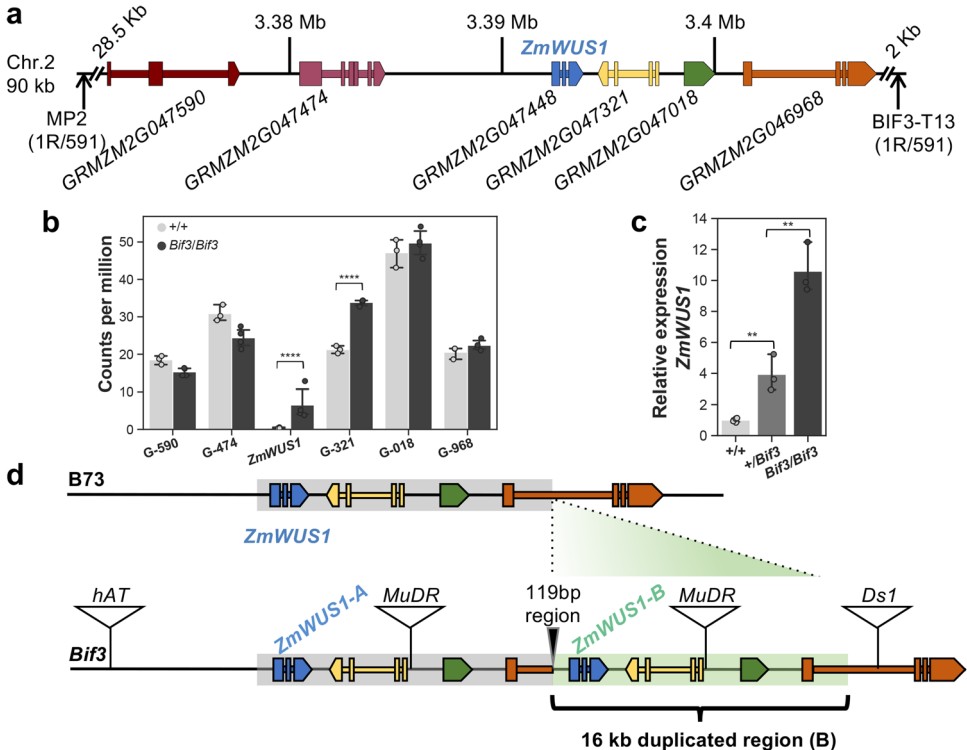

**Fig. 2 Bif3 gene identification. a** Map-based cloning of *Bif3* (in parenthesis, number of recombinants R/total chromosomes). **b** RNA-seq analysis of genes (G) in the mapping window indicates that *ZmWUS1* is differentially expressed (fold-change = 13.6); two-tailed Student's *t*-test ****$p < 0.0001$. $n_{+/+} = 3$, $n_{Bif3/Bif3} = 4$ (*n* pools of three ear samples). Plotted data represent mean ± standard deviation. **c** qRT-PCR of *ZmWUS1* in immature ears; two-tailed Student's *t*-test **$p < 0.01$; $n_{+/+} = 4$, $n_{+/Bif3} = 3$, $n_{Bif3/Bif3} = 3$ (*n* pools of three ear samples). Plotted data represent mean ± standard deviation. **d** Structure of the tandem-duplicated *ZmWUS1* locus in the *Bif3* mutant. A 16 kb fragment containing *ZmWUS1* was duplicated and inserted in the first intron of GRMZM2G046968. Triangles represent transposon insertions from different families.

region (Fig. 3a and Supplementary Figs. 2 and 4), presumably by intrachromosomal recombination-based mechanisms[30].

To unequivocally show that the loss of *ZmWUS1-B* function rescued the mutant phenotype, we used a CRISPR-Cas9 strategy to knock out *ZmWUS1* in *Bif3* mutants. A transgenic construct containing a gRNA targeting the first exon of the *ZmWUS1* coding sequence was first introduced into wild-type maize, and then the resulting line was crossed to a *Bif3* mutant (Fig. 3b). One heterozygous *Bif3* plant resulting from this cross showed a normal inflorescence phenotype (Fig. 3c). By gene-specific amplification and direct sequencing we determined that *ZmWUS1-B* carried a 42 bp deletion/11 bp insertion (CR1) in this plant (Fig. 3c). *ZmWUS1-A* was also edited in this plant, as expected given that the gRNA could not distinguish between the two identical copies of *ZmWUS1*, and carried a 1 bp deletion in the targeted region. Additional editing events (CR2, CR3) were subsequently obtained and produced +/*Bif3* plants with a wild-type phenotype and edited *ZmWUS1-A* and *ZmWUS1-B* copies (Supplementary Fig. 5). Altogether, the two intragenic suppressor approaches demonstrated that the *Bif3* phenotype is caused by the presence of an additional functional copy of the *ZmWUS1* gene.

**Ectopic expression of ZmWUS1-B is responsible for the Bif3 phenotype.** It remained to be determined whether the over-expression of *ZmWUS1* was caused by the presence of an extra copy of the *ZmWUS1* gene, or by the unique *ZmWUS1-B* copy, as suggested by the normal phenotype of the *Bif3-Rev2* individual. To confirm this, we introduced a 10 kb DNA fragment containing the entire *ZmWUS1-A* locus of *Bif3* mutants into wild-type maize,

and found that the plants were significantly shorter than non-transgenic siblings with wider and shorter leaves. However, both tassel and ear developed normally (Supplementary Fig. 6), indicating that adding an extra copy of *ZmWUS1* was not sufficient to phenocopy the inflorescence phenotype of *Bif3* mutants.

Since both *Bif3-Rev1* and *Bif3-Rev2* had normal inflorescences despite harboring both *ZmWUS1-A* and *ZmWUS1-B* copies, we used these revertants to compare the expression levels of *ZmWUS1* by qRT-PCR. While homozygous *Bif3-Rev1* retained elevated expression of *ZmWUS1* (~11-fold), homozygous *Bif3-Rev2* showed only a moderate (~2-fold) up-regulation compared to wild-type siblings (Fig. 3e). Next, we asked whether the *ZmWUS1-B* gene was responsible for the high levels of *ZmWUS1* expression observed in *Bif3* and *Bif3-Rev1*. We took advantage of the G > A SNP in the start codon of *ZmWUS1-B* to design allele-specific primers to quantify the relative contribution of *ZmWUS1-A* and *ZmWUS1-B* transcripts in *Bif3-Rev1*. By qRT-PCR we showed that *ZmWUS1-B* transcripts outnumbered those of *ZmWUS1-A* by approximately 12-fold (Fig. 3f). These results, together with the transgenic approach presented above, suggested that the *Bif3* phenotype was caused by the duplicated *ZmWUS1-B* locus, whose novel chimeric promoter enhanced transcription.

We next analyzed the regulatory sequences upstream of *ZmWUS1-B*, focusing on the unique 119 bp region present in *Bif3* mutants but lost in the *Bif3-Rev2* revertant. We noticed that a 17 bp element, originally present in the *ZmWUS1* promoter in single copy, was repeated three times in this region (Fig. 3g). Within each repeat, we identified an AGATAT motif which is an in vivo binding element of type-B ARR TFs (ARR1, ARR10, ARR12) that activates *WUS* expression in a cytokinin-dependent fashion[15,17,19,20]. This motif of the *WUS* promoter is highly

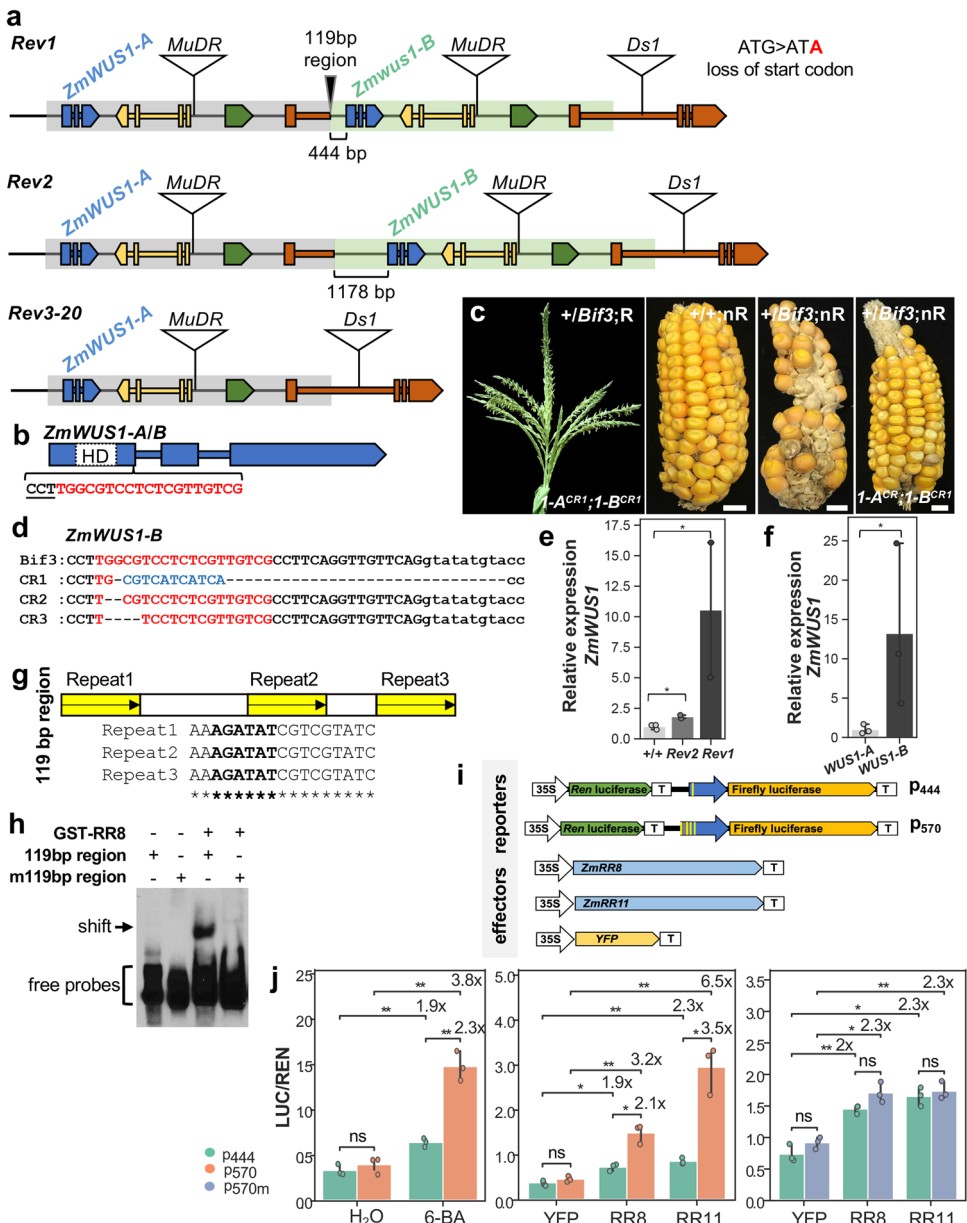

**Fig. 3 Bif3 is caused by ectopic expression of ZmWUS1-B. a** *Bif3* revertants identified from a targeted EMS suppressor screen. Three classes of revertants were identified. Revertant1 (*Rev1*) carries a G-to-A transition occurred at the start codon ATG of *ZmWUS1-B*. In revertant2 (*Rev2*), *ZmWUS1-B* carries a longer promoter (1178 bp from ATG) and lost the 119 bp region. In other revertants (*Rev3–20*), *ZmWUS1-B* is missing. **b–d** CRISPR-Cas9 targeted *ZmWUS1* knock-out in *Bif3* mutant. **b** *ZmWUS1* gene structure with homeobox domain (HD in dashed white rectangle) highlighted in the first exon. gRNA in red with PAM site underlined. **c** The tassel (F1) and ear (BC1) of *ZmWUS1-B* knock-out plants (A619 background). nR, non-resistant to herbicide treatment, R, resistant to herbicide treatment. Scale bars: 1 cm. **d** CRISPR-Cas9 edits in *ZmWUS1-B*. gRNA targeted sequence in red. In blue, insertion sequence. **e**, **f** qRT-PCR of *ZmWUS1* transcripts in wild-type, *Rev2*, and *Rev1* (**e**) and of *ZmWUS1-A* and *ZmWUS1-B* in *Rev1* by copy-specific primers based on the G/A SNP (**f**). Two-tailed Student's *t*-test *$p < 0.05$. $n_{+/+} = 3$, $n_{REV1} = 2$, $n_{REV2} = 2$, $n_{WUS1-A} = 3$, $n_{WUS1-B} = 3$ (*n* independent experiments). **g** The unique 119 bp region of *ZmWUS1-B* contains three direct repeats. Yellow boxes with arrow indicate repeat units, and each repeat unit contains the AGATAT motif. **h** Electrophoretic mobility shift assay (EMSA) with ZmRR8. m119 bp region carries mutations in all repeats. **i**, **j** Transient transactivation in maize protoplasts. **i** Reporter and effector constructs used. The *Firefly luciferase* is driven by the proximal promoter of *ZmWUS1-B* (blue arrows), including 444 bp ($p_{444}$) or 570 bp ($p_{570}$) including the 119 bp region. The *Renilla* (*Ren*) *luciferase* is used for normalization. **j** *ZmWUS1-B* is induced by cytokinin treatments (6-BA), and by type-B ZmRR8 and ZmRR11. Two-tailed Student's *t*-test *$p < 0.05$; **$p < 0.01$. $p_{570m}$, 570 bp proximal promoter with AGATAT motifs mutated into TTTTTT. Plotted data represent mean ± standard deviation. Three replicates for each experiment.

conserved across multiple grass species (Supplementary Fig. 7). We therefore asked if the multimerization of this 6 bp motif caused the up-regulation of *ZmWUS1-B* expression. To this end, we tested whether maize *ZmRR8* and *ZmRR11*, two broadly expressed genes (Supplementary Fig. 7) closely related to Arabidopsis *ARR1*, *ARR10*, *ARR12*, encoded type-B RRs capable

of binding to the *ZmWUS1* promoter region containing the AGATAT motif and activated its transcription. In electrophoretic mobility shift assay (EMSA), ZmRR8 binding was specific and was abolished when the AGATAT motifs were changed to TTTTTT (Fig. 3h). Subsequently, we used a transactivation dual-luciferase transient assay in maize protoplasts and determined

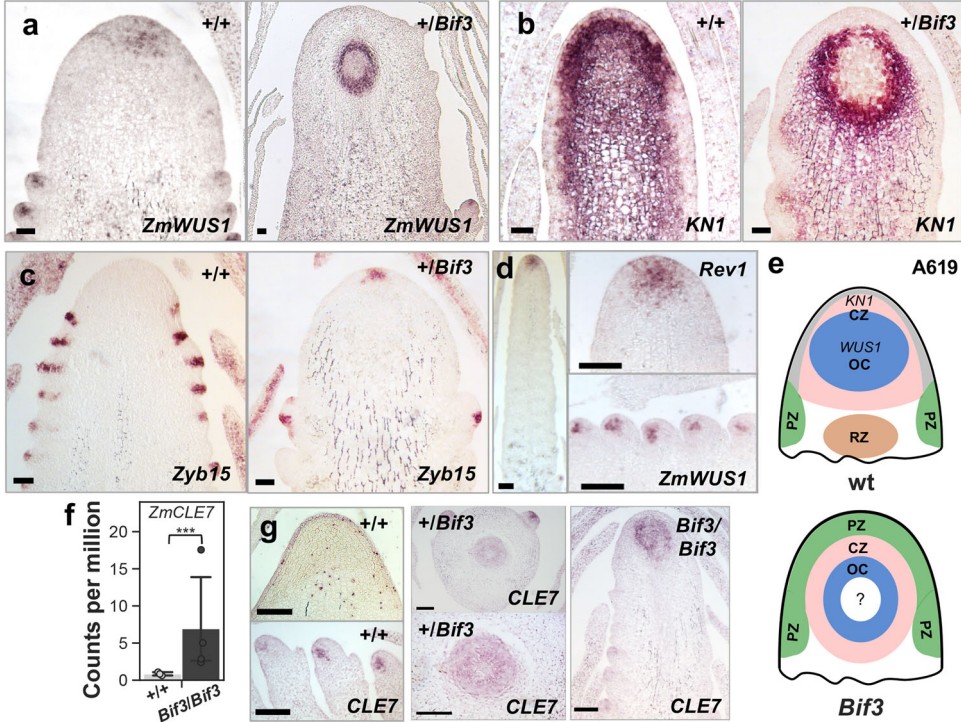

**Fig. 4 *Bif3* architectural rearrangements in inflorescence meristems. a–d** RNA in situ hybridizations of *ZmWUS1* (**a**, **d**), *KN1* (**b**) and *ZYB15* (**c**) in *Bif3* ear inflorescence meristems (IMs). **e** Model of *Bif3* IM structure. OC, organizing center; CZ, central zone; PZ, peripheral zone; RZ, rib zone. **f** Up-regulation of *ZmCLE7* in *Bif3* mutant RNA-seq data. Two-tailed Student's *t*-test ***$p < 0.001$; plotted data represent mean ± standard deviation; $n_{+/+} = 3$, $n_{Bif3/Bif3} = 4$ (*n* pools of three ear samples). **g** Ectopic expression of *ZmCLE7* is observed by in situ hybridizations on longitudinal ($+/+$ and *Bif3/Bif3*) and cross ($+/Bif3$) sections of ear IMs. Scale bars: 50 μm in (**a–c**); 100 μm in (**d**, **g**).

that a 570 bp fragment of the *ZmWUS1-B* promoter containing multimerized copies of the 17 bp sequence ($p_{570}$) had stronger transcriptional activity upon treatment with 10 μM cytokinin (6-Benzylaminopurine, 2.3×, $p < 0.01$) as well as in the presence of overexpressed ZmRR8 and ZmRR11 (2.1× and 3.5×, respectively, $p < 0.05$), than the proximal 444 bp promoter containing a single AGATAT motif ($p_{444}$). Importantly, the latter was not observed when the AGATAT motifs were mutated ($p_{570m}$, Fig. 3i, j). Overall, these results suggest that multiple copies of the AGATAT motif present in the unique 119 bp region of the *ZmWUS1-B* promoter cause its overexpression, and are bound by ZmRRs, making *ZmWUS1-B* hyper-responsive to cytokinin.

**Ectopic expression of *ZmWUS1* causes major architectural rearrangements of inflorescence meristems and mis-regulation of key stem cell regulators.** To determine how AM initiation in *Bif3* inflorescences was disrupted and whether the novel chimeric promoter changed the pattern of expression of *ZmWUS1*, we used RNA in situ hybridizations in immature ears using different marker genes including *ZmWUS1*. In wild-type ears, very weak *ZmWUS1* expression was observed in IMs, while a stronger signal was detected in AMs as previously reported[8] (Fig. 4a). Surprisingly, however, a strong expression of *ZmWUS1* was observed in *Bif3* in a ring-like pattern located deeply in the inflorescence (Fig. 4a). Consecutive sections of this unusual domain showed that it roughly corresponded to a hollow sphere (Supplementary Fig. 8). This pattern was also detected with *KNOTTED1* (*KN1*; Fig. 4b), a meristem marker gene normally expressed throughout the meristem and ground tissue, but excluded from the incipient primordia and the L1 layer[31]. Notably, *ZYB15*, a marker of suppressed bract primordia initiated at the PZ of IMs, was expressed at the tip of the *Bif3* inflorescence structure (Fig. 4c). A

stronger expression of *ZmWUS1* was also observed in homozygous *Bif3-Rev1* immature ears (Fig. 4d), but not in a ring-like domain. Collectively, these data indicated that the ectopic expression of a functional *ZmWUS1-B* copy caused a major rearrangement in IMs giving rise to a meristem with PZ, CZ, OC, and possibly RZ organized in concentric domains (Fig. 4e). This reorganization of the IM could explain the unusual formation of AMs in the tip of the inflorescences, observed by SEMs, and the ball-shaped ears, due to lack of longitudinal growth by the enclosed RZ.

To better understand the consequences of the *Bif3* IM rearrangement, we analyzed the transcriptional changes caused by the *Bif3* mutation in ear primordia, focusing on genes involved in meristem size regulation and hormonal pathways, in particular auxin and cytokinin, critical for meristem function[2]. Differential gene expression analysis from RNA-seq experiments identified 1380 up-regulated and 242 down-regulated genes ($p < 0.05$, FDR $< 0.1$, fold-change $> 1.5$; Supplementary Data 1). Since WUS is a bifunctional transcription factor[32], we performed Gene Ontology (GO) enrichment analysis of genes up- and down-regulated in *Bif3* mutant ears (Fig. 5a). We found strong enrichment for "transcriptional regulation" in up-regulation and down-regulation gene sets, respectively (Fig. 5a). Up-regulated genes included those involved in inflorescence and flower development such as *TGA1* and *ZAG1*[33,34] (Supplementary Data 1). Significantly down-regulated genes included those involved in AM initiation and maintenance including *BAF1*, *BA1*, *RA2*, *RA3*, and *WAB1*[35–39] (Fig. 5b), correlating with failure to initiate AMs in regularly arranged patterns.

We then asked how the *Bif3* mutation affected the CLV-WUS feedback-loop pathway. WUS is sufficient to induce the expression of *CLV3*[14], and RNA-seq data showed that *ZmCLE7*, the likely maize *CLV3* orthologue (*GRMZM2G372364*)[23,40] was

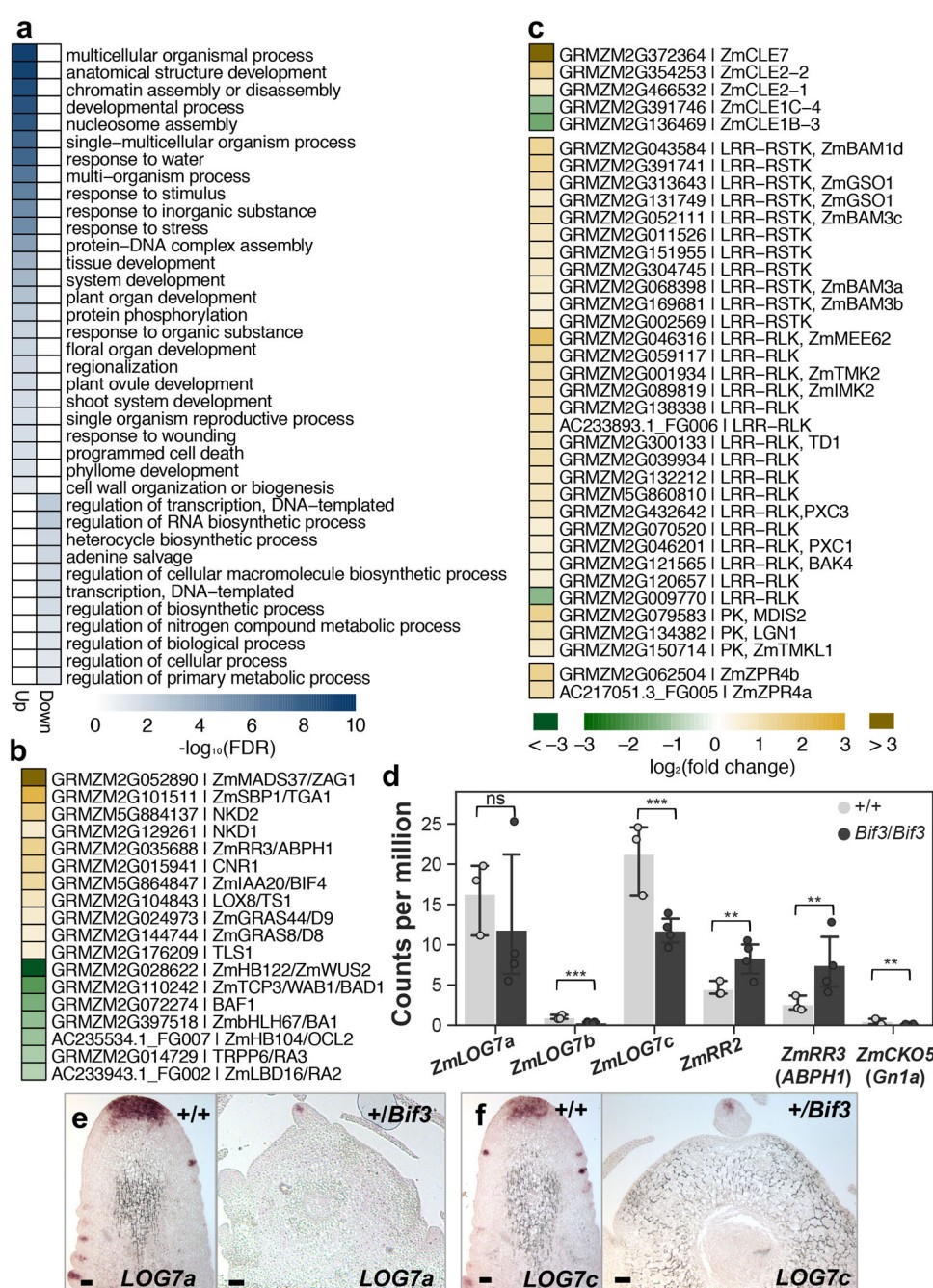

**Fig. 5 Altered expression of key meristem genes in *Bif3* ear primordia. a** GO enrichment analysis of differentially expressed genes. **b** Differential expression of known developmental regulators. **c** Differential expression of CLAVATA3/ESR-related (CLE), protein kinase (PK), leucine-rich repeat (LRR)-receptor-like protein (LRR-RLP), LRR-receptor-like kinase (LRR-RLK), LRR-receptor-like serine/threonine-protein kinase-encoding genes (LRR-RSTK), and LITTLE ZIPPER4 (fold change ≥ 1.5). **d** Gene expression profiles. Two-tailed Student's *t*-test **$p < 0.01$; ***$p < 0.001$; ns, non-significant; plotted data represent mean ± standard deviation; $n_{+/+} = 3$, $n_{Bif3/Bif3} = 4$ (*n* pools of three ear samples). **e**, **f** RNA in situ hybridizations of *ZmLOG7a* and *ZmLOG7c* in *Bif3* inflorescence meristems. Bars: 200 μm.

strongly up-regulated (~8.2-fold) in *Bif3* ear primordia, compared to wild-type siblings (Fig. 4f). In situ hybridization showed that *ZmCLE7*, normally expressed in the meristem L1 layer in IMs and in the CZ of AMs, was ectopically expressed in *Bif3* immature inflorescences in a similar fashion as *ZmWUS1* (Fig. 4f, g). Additional up-regulated genes involved in maize meristem maintenance and the CLV-WUS pathway included two additional *CLE* genes, *ZmCLE2-1* and *ZmCLE2-2* (~1.8 to 2.8-fold), *LIGULELESS NARROW1* (*LGN1*)[41], *THICK TASSEL*

*DWARF1* (*TD1*)[25], and two additional genes encoding protein kinases (PKs), and the paralogous *ZmBAM1d* and *ZmBAM3a* to *3c*, the maize orthologs of the Arabidopsis *BARELY ANY MERISTEM* (*BAM*) genes[42,43]. Furthermore, 14 LRR-receptor-like kinase-encoding genes (LRR-RLK), seven LRR-receptor-like serine/threonine-protein kinase (LRR-RSTK)-encoding genes, and two *LITTLE ZIPPER4* co-orthologs (*ZmZPR4a* and *ZmZPR4b*)[44] (Fig. 5c) were also up-regulated. Notably, among *CLE* genes, *ZmCLE1c-4* and *ZmCLE1b-3* were instead down-

regulated (2.2-fold and 3.1-fold, respectively) in *Bif3* ear primordia (Fig. 5c).

Since cytokinin is an essential hormone for meristem maintenance, we examined the expression of the *LONELY GUY* (*LOG*)[45] maize co-orthologs. *LOG* encodes a cytokinin-activating enzyme in the final biosynthetic step to produce free-base cytokinin to maintain meristem activity. *LOG* is expressed in the CZ of IMs[45] in rice, and two of the three maize *LOG* co-orthologs[46] were expressed in the CZ of IMs and AMs of wild-type ears. These two genes were down-regulated in *Bif3* ear primordia (Fig. 5d), and were expressed only in the AMs of *Bif3* ears and not in IMs (Fig. 5e, f). Additionally, we determined that two type-A *ZmRR* genes, *ZmRR2* and *ZmRR3/ABPH1*, which act as negative feedback regulators of cytokinin function, were significantly up-regulated[47]. All these data suggested that cytokinin biosynthesis and response are severely impaired in *Bif3* IMs.

Together with cytokinin, auxin is a key hormone controlling basic meristem function[2]. We therefore checked the interaction of *BIF3* with *BIF4*, which encodes an Aux/IAA protein[48] that was significantly up-regulated in *Bif3* ear primordia (Supplementary Fig. 9). We tested the genetic interaction of *Bif3* and *Bif4*, a semi-dominant inflorescence mutant that alters branch and spikelet formation, in the B73 background, which mitigates the severity of the heterozygous *Bif3* phenotype (Supplementary Fig. 1). SEMs of tassel and ear primordia revealed that heterozygous *Bif3* inflorescences in B73 had normal IMs while only ears had patterning defects. However, both tassels and ears had enlarged IMs and were severely affected in inflorescence patterning in homozygous *Bif3* mutants (Supplementary Fig. 9). Surprisingly, in situ hybridizations with *ZmWUS1* and *KN1* in *Bif3/Bif3* ears often revealed two OCs (Supplementary Fig. 9), suggesting that immature mutant ears tried to recover and form a normal IM. Double heterozygous +/*Bif3*;+/*Bif4* mutants dramatically enhanced the phenotype of single +/*Bif3* or +/*Bif4* phenotypes, with ears showing extensive barren patches (Supplementary Figs. 1 and 9). SEM analysis of +/*Bif3*;+/*Bif4* immature ears showed disorganized IMs with few irregularly arranged AMs, when compared to both single mutants (Supplementary Figs. 1 and 9). By in situ hybridization we determined that *ZmWUS1* expression was still up-regulated in the IM of +/*Bif3*;+/*Bif4* double mutants (Supplementary Fig. 9). Furthermore, the expression of *KN1* was excluded from an additional cell layer of +/*Bif3* and +/*Bif3*;+/*Bif4* IMs when compared to wild-type and +/*Bif4* inflorescences (Fig. 4b and Supplementary Fig. 9), and *ZmLOG7a* in the CZ of +/*Bif3*;+/*Bif4* IMs was missing (Supplementary Fig. 9). Altogether, elevated *ZmWUS1* expression caused disruption of auxin and cytokinin pathways, leading to major architectural rearrangements of IM and defects in initiation and maintenance of AM in *Bif3* ears.

**Fasciated ear mutants enhance *Bif3* inflorescence defects.** *WUS* expression is controlled by the CLV-WUS negative feedback-loop which in maize includes the CLV2 ortholog FEA2, the LRR-receptor-like protein FEA3, and the Gα subunit of heterotrimeric GTP binding protein CT2[21,23,24], among others. However, the function of *WUS* in maize remains unresolved. We therefore investigated the genetic interactions between *Bif3* and fasciated mutants using the less severe *Bif3* mutant in the B73 background for double mutant analysis.

*Bif3* dramatically enhanced the phenotype of single *fea2*, *fea3*, and *ct2* mutants, showing in ears more enlarged and fasciated IMs, stronger than those observed in single *Bif3* IMs (Fig. 6a, d, g and Supplementary Fig. 10). The enhancement was also evident in mature tassels, in particular in double mutant combinations

with *ct2* and *fea3* that showed extremely reduced and barren central spikes (Supplementary Fig. 10). These results indicated that removing negative regulation of *ZmWUS1* expression enhanced the *Bif3* phenotype, consistent with a model in which FEA2, FEA3 and CT2 negatively regulate *ZmWUS1* expression to maintain IMs. We also examined the expression patterns of *KN1* and *ZmWUS1* in double mutant IMs. Double *fea2/fea2*;*Bif3/Bif3* and *ct2/ct2*;*Bif3/Bif3* mutants retained the ring-shaped expression of *ZmWUS1* and *KN1* in extremely enlarged IMs. Intriguingly, this expression pattern was not detected in the *fea3/fea3*;*Bif3/Bif3*, and the IM split in multiple apical meristems (Fig. 6g, h).

FEA2 and CT2 are receptor components that participate in CLE7 signaling, while FEA3 perceives the ZmFCP1 peptide in a parallel signaling pathway to the CLV-WUS pathway[6,23]. FEA3 functions to restrict *ZmWUS1* expression and exclude it from the RZ[23,46]. We therefore tested the sensitivity of *Bif3* embryonic SAMs to both ZmCLE7 and ZmFCP1 in comparison to wild-type embryos. As expected due to *ZmWUS1* overexpression (Supplementary Fig. 10), +/*Bif3* SAMs showed partial resistance to both peptides as they were less sensitive to the treatments than wild-type embryos (Fig. 6j, k). Altogether, these findings suggest that in maize, unlike rice, the function of WUS in the CLV-WUS feedback pathway is conserved.

## Discussion

Recent tandem duplications are the most challenging structural variations to identify in genomes and have been significant drivers of phenotypic variability in traits important for domestication and breeding of crop species[49–51]. Here we determined that the *ZmWUS1* locus underwent a large tandem segmental duplication event in *Bif3* mutants that created a unique 119 bp insertion in the proximal promoter region of the duplicate *ZmWUS1-B* copy. Segmental duplications are a typical feature of many plant and animal genomes[52,53], and are known to generate new genes, diversify gene functions, and expand gene families in the course of genome evolution[54]. The new copies of the duplicated genes can acquire modified functions compared to the ancestral ones, as in the case of the maize dominant pod corn mutant *Tunicate1*, which is caused by a long-range chromosomal inversion followed by a tandem duplication of the MADS box transcription factor *Zmm19* in the junction region[55]. However, long chromosome segment repeats are often unstable and lead to repeat losses, as observed for example in other dominant maize mutants[56,57]. In *Bif3* mutants, instability of the tandem-duplicated structure was obvious during the mutagenesis screen and since *MuDR* has been reported to be active under these conditions[58], it may have been triggered by the pollen EMS treatment.

While the CLV-WUS pathway has been known for decades, an open and long-standing question is whether the function of individual members of the pathway is conserved across plant species. Notably, the rice *WOX4* gene provides an analogous function to *WUS* in meristems, while the true *WUS* ortholog *TAB1* has only a role in AM initiation[9,11]. Our data in maize indicate that the *ZmWUS1* is involved in both meristem maintenance and AM initiation. However, we cannot rule out that the defects in AM initiation observed in *Bif3* mutant inflorescences may be a consequence of the drastic reorganization of IMs, whereby the CZ, OC and possibly the RZ are deeply embedded in differentiated tissue. Hence the signal that normally triggers the formation of lateral primordia has to travel through several more cell layers to reach the surface of the inflorescence structure and this could be the cause of the observed patterning defects. Nonetheless, our genetic and molecular analysis of the interaction of ZmWUS1 and the various components of the CLV-WUS pathway are consistent with conservation of the pathway in

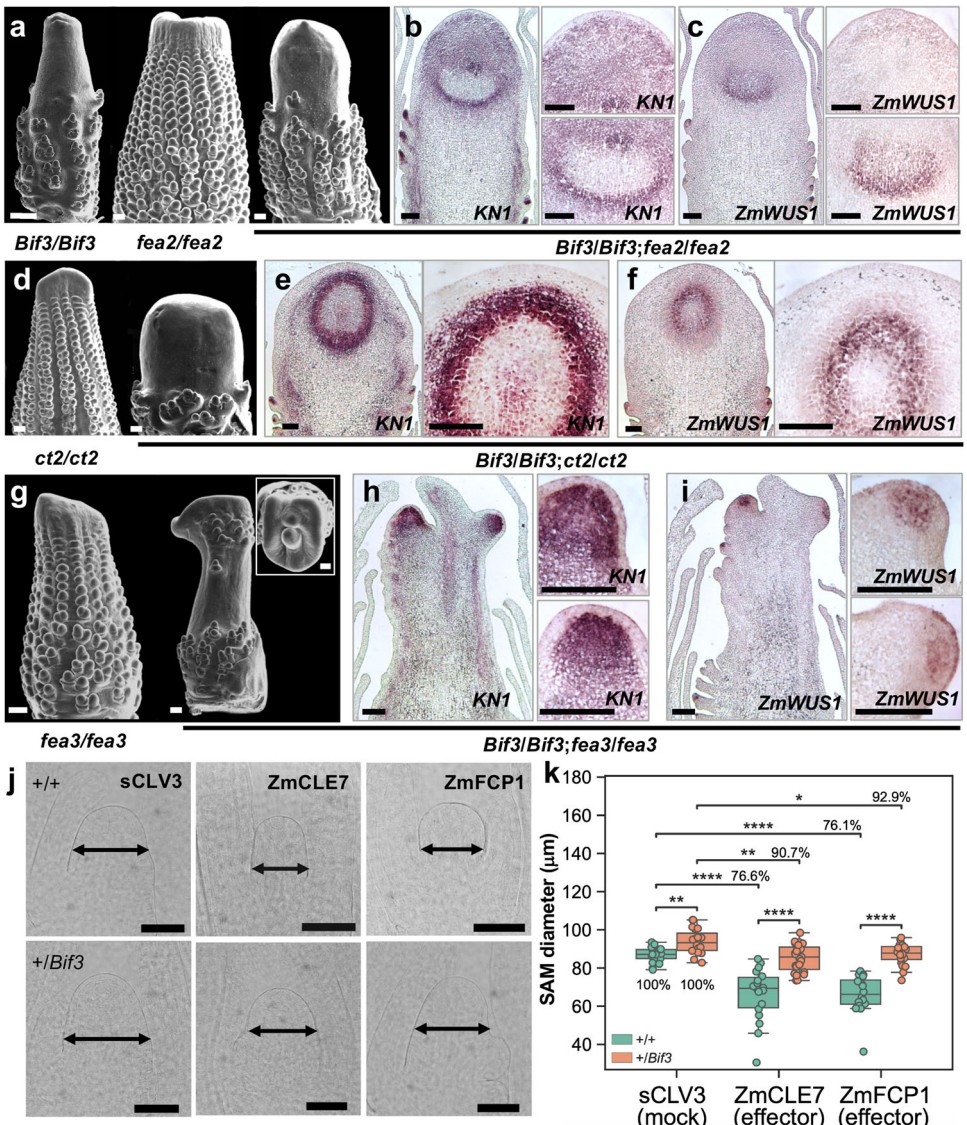

**Fig. 6 Genetic and molecular interaction of *Bif3* and negative regulators of *ZmWUS1*. a, d, g** SEMs of single and double mutant ears in B73 background. Scale bars: 200 μm. **b, c, e, f, h, i** In situ hybridizations in double mutant immature ears with *ZmWUS1* and *KN1* antisense probes. Scale bars: 50 μm. **j, k** *Bif3* SAMs are partially resistant to ZmCLE7 and ZmFCP1 peptide treatments. Percentages in (**k**) indicate reduction relative to scrambled peptide control (sCLV3). Two-tailed Student's *t*-test \*$p < 0.05$; \*\*$p < 0.01$; \*\*\*$p < 0.001$; \*\*\*\*$p < 0.0001$ ($n_{+/+\ sCLV3} = 17$; $n_{+/Bif3\ sCLV3} = 19$; $n_{+/+\ ZmCLE7} = 18$; $n_{+/Bif3\ ZmCLE7} = 23$; $n_{+/+\ ZmFCP1} = 16$; $n_{+/Bif3\ ZmFCP1} = 20$). Box plot center line, median; box limits, upper and lower quartiles; whiskers, maximum and minimum values. Single datapoints outside of the whiskers represent outliers Scale bars: 100 μm.

maize, namely that *ZmWUS1* functions in meristem size regulation, suggesting that rice may not be representative of all monocots. In support of its role in meristem size regulation, our data show that *ZmWUS1* is expressed at a very low level in IMs and agree with data from a published transcriptional reporter line[23]. Our transgenic analysis also showed that increasing *ZmWUS1* expression by altering copy number (Supplementary Fig. 6) has consequences to vegetative development, suggesting that tight transcriptional control of *ZmWUS1* is essential for SAM function. Finally, we conclude that CRISPR-induced mutations in *ZmWUS1* do not appear to have phenotypic consequences for overall maize development (Supplementary Fig. 9), likely due to the presence of the paralogous *ZmWUS2* gene[8], suggesting that the two genes play redundant functions in meristem development. Alternatively, other *WOX* genes could work redundantly with *ZmWUS1*. Recently, an enlarged ball-shaped SAM was observed in Arabidopsis plants expressing a constitutively active

version of a type-B ARR[20] which mimics constant cytokinin response. This is consistent with our findings and indicates that a similar rearrangement of the meristem structure occurred in these Arabidopsis plants and raises intriguing questions on why hypersensitivity to cytokinin by *WUS* genes drives the formation of spherical meristems.

Tinkering with meristem size holds promise for increasing yield in different plant species[26,59,60]. The dominant *Bif3* mutation places *ZmWUS1* squarely at the center of meristem size regulation in maize and provides new insights into the transcriptional control of *WUS* genes. Significant interest has been placed on the function of *ZmWUS* genes given their use alone or in combination with the transcription factor BBM for improving transformation of maize and other recalcitrant species[12,13]. Our results indicate that high levels of expression of *ZmWUS1* can be achieved by multimerization of a single highly conserved cis-regulatory element of the proximal promoter sequence. These

findings could be applied to different species to boost the expression of morphogenetic factors in transformation systems.

## Methods

**Plant materials and genotyping.** The original *Bif3* allele (*Bif3-ref*) was a donation by Matt Evans and Paula McSteen. All mutants used for phenotypic and molecular analysis were backcrossed to the A619 (BC10) and B73 (BC8) inbred lines. For double mutant analysis in the B73 background, we used the *fea2-0*, *fea3-0*, *ct2-ref*, and *Bif4-N2616* alleles from previous reports[6,23,24,48]. Double mutant genotype was determined using allele-specific primers (Supplementary Data 2).

**Plant phenotyping.** Images of IMs were captured by scanning electron microscopy (SEM) while those of SAMs were captured on FAA-fixed and methyl salicylate-cleared seedling tissues by differential interference contrast (DIC) microscopy. For SEMs, fresh tassel and ear primordia (3–5 mm) were dissected and directly analyzed using a JMC-6000PLUS benchtop SEM. For quantifying meristem width, 7-day-old siblings containing SAMs were dissected and fixed in FAA (2% formaldehyde, 5% acetic acid, 60% ethanol) overnight at 4 °C. The fixed tissues were dehydrated in a graded ethanol series (75%, 85%, 95%, and 100%) for 30 min each at room temperature and cleared in methyl salicylate/ethanol series (50% and 100%) for 1 h each at room temperature, followed by 100% methyl salicylate overnight treatment. Meristems were mounted in methyl salicylate on slides with coverslips and imaged using a LEICA DM5500B microscope. Measurements were taken using Fiji (ImageJ); the width of the meristem was calculated using P1 as a reference.

Leaf width (approximately 15 cm above ligule) and length (leaf tip to ligule) of +/*Bif3* mutants and *pZmWUS1-A::ZmWUS1-A* transgenic lines were measured on the top-third leaf blades.

**EMS mutagenesis and screen.** For EMS mutagenesis screen, pollen of A619 *Bif3/Bif3* mutants was collected in the morning, treated with EMS (0.06% in mineral oil) for 45 min and used to fertilize A619 +/+ ears. Seeds of the resulting M1 progeny were planted in the field in two consecutive years and revertants were identified based on tassel and ear phenotypes. For the 3D-pooling strategy, we first pooled equal amounts of leaf tissues, extracted genomic DNA, and amplified the coding sequence of *ZmWUS1* (both *ZmWUS1-A* and *ZmWUS1-B* copies) by PCR using primers WUSseqF3 and WUSseqR3. Subsequently, the PCR products were purified using QIAquick PCR Purification Kit (QIAGEN) and sonicated to 200 bp fragments in a Covaris S2 sonicator. DNA fragments were subsequently purified using AmpureXP beads at a 2:1 bead:DNA ratio. The fragmented DNA was then end-repaired using the End-It kit (Lucigen) followed by DNA purification using QIAquick PCR Purification Kit (QIAGEN). The purified samples were then A-tailed using Klenow Fragment for 30 min at room temperature and again purified using QIAquick PCR Purification (QIAGEN). The A-tailed DNA fragments were then ligated overnight with a truncated Illumina Y-adapter[61]. Finally, these DNA libraries with unique barcodes were purified by bead cleaning using a 1:1 bead:DNA ratio, and then eluted from the beads in 30 µL of elution buffer, followed by double-stranded DNA quantification using the Qubit dsDNA HS Assay Kit. DNA libraries were pooled and sequenced on an Illumina NextSeq500 sequencer. Reads were mapped to *ZmWUS1* of *Bif3* allele using software UGENE[62], and the alignment results and the nucleotide changes were detected using Integrative Genomics Viewer (IGV)[63].

**Positional cloning of *Bif3*.** The *Bif3* locus was initially mapped to chromosome 2 between bins 2.00 and 2.01 by bulk segregant analysis (BSA) using a +/*Bif3* × Mo17 BC₁ population[64]. Fine mapping of the *Bif3* locus was obtained using molecular markers available from MaizeGDB[65] on approximately 590 mutant individuals from a +/*Bif3* × Mo17 BC₂ population. Molecular markers are listed in Supplementary Data 2.

The tandem duplication in the *Bif3* locus was determined by Southern blotting and thermal asymmetric interlaced polymerase chain reaction (TAIL-PCR). For Southern blotting, genomic DNA from *Bif3* homozygous mutant and wild-type siblings was extracted from leaves. 10µg of genomic DNA was digested with restriction enzymes (*Eco*RI, *Xba*I, *Hind*III, and *Pst*I) at 37 °C for 6 h. The fragmented DNA was resolved by size on a 0.7% agarose gel, denatured, and blotted onto a Hybond-N membrane (GE Healthcare). The membrane was cross-linked by UV light and hybridized with *ZmWUS1*-specific DNA probe. This ~1 kb *ZmWUS1*-specific DNA probe spanned from the proximal promoter to the second intron (from −335 bp to +708 bp) and was amplified using primers WUSseqF3 and WUSfu-R1, and then cloned into pGEM T-easy vector. The DNA probe was isotope-labeled with Random Primer DNA Labeling Kit Ver.2 (TaKaRa) and purified with ProbeQuant G-50 Micro Column (GE Healthcare). For *Bif3* TAIL-PCR chromosome walking, symmetric and asymmetric PCRs were performed as described previously[27]; the amplicons that ranged from 2 to 6 kb were recovered from gel and cloned into pCR-Blunt vector (Thermo Fisher Scientific) for sequencing. Primers are listed in Supplementary Data 2.

Multiple sequence alignments were performed using Clustal Omega (https://www.ebi.ac.uk/Tools/msa/clustalo/).

**Generation of maize transgenic lines.** Transgenic maize plants carrying the *pZmWUS1-A::ZmWUS1-A* construct and CRISPR-Cas9-derived mutants and their wild-type segregants were grown under standard long-day greenhouse conditions (16 h light and 8 h dark) or in the field. To obtain the CRISPR-Cas9-derived mutants, a *ZmWUS1*-specific guide RNA was cloned into pBUE411 vector by Golden Gate cloning, and the resulting construct was introduced into Hi-II immature embryos by Agrobacterium-mediated transformation, followed by callus induction, bialaphos selection, and shoot and root regeneration[66]. CRISPR-Cas9-derived *ZmWUS1-B* knock-out mutation in *Bif3* mutant was created by crossing transgenic lines carrying CRISPR-Cas9-*ZmWUS1*gRNA construct with *Bif3* mutants. The genotype of the knock-out mutations was determined by copy-specific genotyping primers—1428WUS-FWD/WUSseqR1 (for *ZmWUS1-A*) 6968-F7/WUSseqR1 (for *ZmWUS1-B*)—followed by direct sequencing using primer Bif3RTF3-1.

The *pZmWUS1-A::ZmWUS1-A* transgenic construct was created by cloning a 10 kb genomic fragment including the 7.7 kb upstream of ATG, the 1.2 kb gene body (ATG to TGA) and the 1.1 kb downstream fragment of TGA of the *Bif3* mutant into the pTF101.1 transformation vector. The resulting construct was introduced into Hi-II maize by Agrobacterium-mediated transformation method at the Iowa State University Plant Transformation Facility. Out of 15 original events, only two events survived and made it to flowering. These two events were crossed with A619 and then backcrossed twice for phenotypic analysis. Genotyping of the transgenic construct was done using primers 7448-F12 and 7448-R9. Additionally, the presence of the transgenic construct was monitored by treatment of leaf blades of 3-week-old plants with 0.4% Liberty®. Primers are listed in Supplementary Data 2.

**Expression and RNA-seq analysis.** In situ hybridization experiments were performed on 2–4 mm ear and tassel primordia fixed in PFA, dehydrated in a graded ethanol series, cleared with Histoclear and embedded in Paraplast[48]. To generate the antisense RNA probes for *ZmWUS1*, *ZmLOG7a*, *ZmLOG7c*, *ZmOCL4*, *ZmIAA9*, *ZYB15*, *KN1*, and *ZmCLE7* genes, the entire/partial coding sequences together with partial 3′-untranslated region of each gene were cloned into pENTR223-Sfi, pBJ36, or pGEM T-easy vectors (Promega). The resulting plasmids were linearized by restriction endonucleases, and the antisense RNA probes (with sizes ranging from 400 to 1000 bp) were synthesized by T7 or SP6 RNA polymerase (Promega). The vectors, enzymes, and primers used for probe design are listed in Supplementary Data 2.

qRT-PCR analysis was performed using an Illumina Eco Real-Time PCR System. At least three ear primordia were pooled for each genotype and two biological replicates were performed for each assay. Total mRNA was extracted using RNeasy Mini Kit (Qiagen) with on-column DNase I (Qiagen) treatment and used for complementary DNA (cDNA) synthesis with a qScript cDNA synthesis kit (Quanta Biosciences). qPCR was performed using gene-specific primers and the PerfeCTa SYBR Green FastMix (Quanta Biosciences). Target cycle threshold values were normalized using *ZmUBIQUITIN*. Primers used are listed in Supplementary Data 2.

For RNA-sequencing analysis, ear primordia (3–4 mm) of homozygous *Bif3* mutant and wild-type siblings in B73 background were dissected and frozen in liquid nitrogen. Total RNA was extracted using RNeasy mini kit (Qiagen) with on-column DNase I (Qiagen) treatment. Four biological replicates for homozygous *Bif3* (three ear primordia for each replicate) and three biological replicates for wild-type (three ear primordia for each replicate) were performed for RNA-sequencing analysis. RNA-seq raw data were trimmed with Trimmomatic with default settings[67] and aligned to the reference genome (B73v3 genome) using HISAT 2.1.0 with default settings[68]. Gene expression was quantified as read counts by HTSeq-count[69] and genes with differential expressions were determined by edgeR 3.18.1 package[70]. All statistical analyses of gene expression were conducted in R and python. Genes with a fold change > 1.5 with $p < 0.05$ and FDR < 0.1 were considered as differentially expressed genes.

**EMSA and transactivation assays.** For EMSAs, the 119 bp repeat of *ZmWUS1-B* novel promoter that contained the putative B-ARR core AGATAT element was amplified from *Bif3* mutant, and biotinylated using the Biotin 3′ End DNA Labeling Kit (ThermoScientific) according to the manufacturer's recommendations. The recombinant protein GST-ZmRR8 (DNA-binding domain only, 191aa-262aa) was generated by cloning DNA-binding domain sequences of ZmRR8 into the pET-60-DEST vector, followed by IPTG (isopropyl β-D-1-thiogalactopyranoside)-induced expression in Rosetta (DE3) cells (Stratagene). Cells from 500 mL Terrific Broth (TB) medium were harvested and resuspended in 10 mL resuspension buffer containing 25 mM Tris-HCl (pH 7.5), 150 mM NaCl, 0.5% Triton X-100, 1 mM DTT, protease inhibitor mixture (Roche), and 2 mM lysozyme and then sonicated on ice. Cleared lysate was applied to GST Gravitrap Glutathione Sepharose 4B (GE Healthcare) columns and washed with resuspension buffer thoroughly. Recombinant proteins were eluted with 10 mM reduced glutathione and concentrated using Amicon Ultra 30 K concentrators (Millipore). DNA binding assays were performed using the Lightshift Chemiluminescent EMSA kit (ThermoScientific) as follows: binding reactions containing 1× Binding Buffer, 50 ng/µL poly(dI/dC), 2.5% glycerol, 2 µL biotinylated probe, and 1 µL purified GST-ZmRR8 protein were incubated at room temperature for 20 min and loaded on a

6% DNA retardation gel (Life Technologies), followed by transferring the DNA probes and proteins to a nylon membrane. Subsequent detection was carried out according to the manufacturer's recommendations. The mutated probe was generated by oligonucleotide synthesis, and the three core AGATAT elements were changed to three TTTTTT elements.

Transactivation assays were conducted in maize mesophyll protoplasts as described previously[71]. The effector vector pEarlyGate 100 was used for expression of ZmRR8, ZmRR11 and YFP driven by the cauliflower mosaic virus (CaMV) dual 35S promoter. The reporter vector pGreenII 0800-LUC was used for detecting the transactivation activities of ZmRR8 and ZmRR11 on the ZmWUS1-B novel promoter with 119 bp insertion ($p_{570}$) and ZmWUS1-B original promoter without 119 bp insertion ($p_{444}$), respectively. Mutated elements in which the three core AGATAT elements of the 119 bp region were changed to TTTTTT by oligonucleotide synthesis (Sigma). Equal amounts of effector plasmids and reporter plasmids were co-transformed into mesophyll protoplasts of maize by PEG-mediated protoplast transformation[72]. The protoplasts were incubated in dark at 23 °C for 15 h for Firefly luciferase/Renilla luciferase (LUC/REN) activity analysis. The ratio of LUC/REN activity was measured using the Dual-Luciferase® Reporter Assay System (Promega). Primers used to amplify the ZmWUS1-B promoter and the coding sequences of ZmRR8 and ZmRR11 are listed in Supplementary Data 2.

**Peptide assays.** Maize embryos segregating for Bif3 and wild-type were dissected 10 days after pollination and cultured on gel medium supplied with scrambled peptide, ZmFCP1 or ZmCLE7 peptide (30 μM each; Genscript) as described previously[6]. After 10 days, embryos were collected for genotyping, fixed and cleared with methyl salicylate as described above. SAMs were then imaged using a LEICA DM5500B microscopy and the SAM width was measured using ImageJ as described above.

**Statistics and reproducibility.** Statistical significance was determined by two-tailed Student's t-tests; exact p values for all comparisons are provided in the Source Data file. All experiments have been replicated at least twice with similar or identical results and/or data have been extracted using multiple biological samples (e.g., RNA-seq, EMS mutagenesis). For in situ hybridizations, at least two independent samples were hybridized with similar results; a representative picture is shown in the figures. Similarly, representative SEMs of the different genotypes analyzed are presented in the figures.

**Reporting summary.** Further information on research design is available in the Nature Research Reporting Summary linked to this article.

## Data availability

RNA-seq datasets are deposited in GEO with accession code GSE158330. Genomic sequences of ZmWUS1-A and ZmWUS1-B loci in the Bif3 mutant are available in GenBank (accession numbers MW677561, MW677562, MW677563). Source data are provided with this paper.

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

## Acknowledgements

We are grateful to Matt Evans and Paula McSteen for donating the original *Bif3* allele; to Dave Jackson and members of the Gallavotti lab for critical reading of the manuscript; to Weibin Song for suggestions on maize transformation; to Caitlin Menello for assistance with revertant characterization and Qiguo Yu for assistance with Southern blots; to Robert Schmitz and the Georgia University Genetic Department for RNA-seq library preparation and sequencing. This work was supported by grants from the National Science Foundation (IOS#1546873 and IOS#2026561) to A.G. and by the Charles and a Johanna Busch Postdoctoral Fellowship to Z.C.

## Author contributions

Z.C., M.G., and A.G. designed research and analyzed data; Z.C., W.L., C.G., A.B., M.G., and A.G. performed experiments; Z.C. and A.G. wrote the paper.

## Competing interests

The authors declare no competing interests.
