## [Peer Review File · Nature Communications]

REVIEWER COMMENTS

Reviewer #1 (Remarks to the Author):

This excellent paper describes an unusual dominant mutation in the maize WUS1 gene. Although the meristem is fasciated, it is also somewhat barren, unlike the other fasciated ear (fae) mutants in the maize CLV pathway. They identify a tandem duplication that includes WUS and a couple other linked genes. They show that it is the WUS gene by analysis of revertants and generating CRISPR alleles in the dominant mutant allele background. One copy is in the wild-type context (WUS1-A) and the other has a new promoter (WUS1-B).

They use EMS to obtain revertants. One revertant is an EMS mutation in the start codon of WUS1-B. Another revertant has a longer promoter in front of WUS1-B, and 17 others have lost a copy of the duplication. They obtain CRISPR lines that were mutated in both copies. These have normal phenotypes.

They analyze the novel promoter and find it has three copies of a region that is normally present in one copy. This sequence is the binding site for type B ARR proteins. Indeed, they show through EMSAs that ZmRR8 binds this region. They carry out a dual luciferase transient assay in maize protoplasts using a 570 piece of the promoter for WUS1-B. It has stronger activity in the presence of CK than the promoter for WUS1-A (with one copy of the binding region). This is a nice experiment.

They also make transgenics that have a second copy of the gene in its native context. The plants are shorter, but the inflorescence is not different.

The in situs are the most intriguing. They find a hollow sphere through the IM that is devoid of signal. The circle of KN1 and WUS expression is bizarre but the fact it is perpendicular to the axis of the inflorescence is really strange. The serial sections in extended data figure 7 are essential to understand this pattern. I also like the image of zyb15 at the very tip of the IM. The LOG7 in situs are also stunning. This paper really demonstrates the importance of looking at the pattern of expression with in situs.

They carry out double mutants with CLV mutants. They start this section (line 336) by first describing the SEMS and in situs of Bif3 in B73. I think they may want to bring these points up in the previous section, or at least start a new paragraph at line 349. It is interesting that fea3 does not behave the same as fea2 or ct2.

Minor points

Line 134. Did they also do qRT-PCR on GRMZM2G047321? Not necessary to do the experiment as they prove it is WUS, but it would be worth noting if they did it.

I am curious about the similarity to Ki3. They point to extended data fig 2, but there isn't much there. Did Ki3 also have a duplication or does it have some of this novel promoter?

They suggest in the discussion that maize can sustain null mutations in WUS1 given their CRISPR alleles, however, did they actually analyze plants that were homozygous for these mutations? It seems the analysis was in the context of one normal copy.

Extended data Fig. 8. It seems the hollow tube is missing in the Bif3/Bif4 double. That is what caught my attention.

Line 372. Don't use the word recent twice in one sentence.

The locus reminds me of Kn1-O, which also has a novel promoter and alleles similar to your Rev2.
Sarah Hake

Reviewer #2 (Remarks to the Author):

The manuscript "Structural variation at the maize WUSCHEL1 locus alters stem cell organization in inflorescences" reports the cloning and characterisation of a novel WUS mutant in maize for the first time. The authors careful, and meticulous confirmation of the causal mutation shows that

genomic rearrangement around the WUS locus caused significant morphological changes in the meristem. This is a well written paper which exhibits a huge quantity of careful investigation of this complex mutant. Their work clearly shows a principal role of WUS gene function in maize meristem regulation in contrast to work in rice, highlighting diversification in meristem regulation across the monocots which could be key in modulation of meristem size to increase yield in different monocot crops.

This is an excellent study and my comments below are for very minor improvements.

- Please include N numbers for the in situ shown
 - Line 118: "As a result, the leaves of Bif3 mutants were 20% wider and 12% shorter than wildtype". I would recommend altering this phrase, as it implies that the authors proved that meristem size has directly altered leaf size.
 - Line 132: further clarification on why the gene downstream of WUS, which was also upregulated, was discounted would be beneficial.
 - Extended Data 2: the text refers of panel A showing wild type, Bif3 mutants and revertants, but the panel appear to only be of wild type and Bif3 mutants.
 - Line 153: "only present in Bif3 mutant backgrounds and not in diverse inbred lines" refers to Extended Data Figure 2, but it is not clear in the figure that multiple inbred lines have been tested. Perhaps reorder the text?
 - Figure 3C: is there an error in R and nR labelling?
 - In several of the figures a more complete family profile would be valuable in helping highlight the differences between the Bif3 mutants, the wild type of second mutant background and the double mutants, as different stages and tissues are shown.
- Line 350: the authors state that the ears are enlarged and more fasciated than in single mutants. Is there a way to quantify this difference?

Reviewer #3 (Remarks to the Author):

The manuscript "STRUCTURAL VARIATION AT THE MAIZE WUSCHEL1 LOCUS ALTERS STEM CELL ORGANIZATION IN INFLORESCENCES" presents a beautiful dataset outlining the functional potency of ZmWus, one of the orthologs of the stem cell regulator WUS.

The manuscript contains a host of genetic, molecular and genomic data to demonstrate that the cause of barren inflorescence3 (Bif3) mutant phenotype lies in the ectopic expression of ZmWUS1. The mechanism for this misexpression is revealed to be a genomic duplication of the ZmWUS locus, along with a multimerization of cytokinin Type-B Response Regulator binding sites. This multiplication of positive regulatory elements is put forward as the cause for WUS transcriptional activation. After having identified the molecular cause of the phenotype they thoroughly characterize the interaction of the mutant with known regulators of maize meristem regulation by genetics and expression studies.

Overall, the manuscript makes a very convincing case to show that ZmWUS1 can affect meristem development and that the phenotype is likely to be driven by cytokinin signalling. The data is of high quality and ample quantity, beautifully presented and well described in the text.

Unfortunately, we do not learn much about the in vivo function of ZmWUS, nor about its regulation. The authors demonstrate with their CRISPR lines that null alleles of ZmWUS1 have no defect, and the data on cytokinin inducibility shows an exclusive effect on the artificial promoter of the mutant. I do recognize that overexpression studies can make important contributions to our understanding of a biological process that are relevant to a wider audience, but in the complete absence of loss-of-function supported functional evidence this does not seem to be the case here. The authors argue that functional redundancy with ZmWUS2 is masking the effect of removing ZmWUS1, but the diverging expression domains of both genes would argue otherwise. Since the authors already have generated null alleles for ZmWUS1, I strongly recommend to use the same strategy to produce ZmWUS2 null mutants for double mutant analysis. I am afraid that without a

clear loss-of-function phenotype for ZmWUS, the manuscript will only appeal to a rather small group of WUS enthusiasts and hence will remain suitable only for much more specialized journals.

Minor comment: The illustration in Fig4 e does not correlate with the WUS expression domain shown in panel a. The data seem to suggest that WUS mRNA can be found at the apical dome, but the illustration suggests otherwise. Please also discuss published data on ZmWUS expression, they do seem to differ.

We wish to thank all reviewers for their helpful comments and suggestions on our previous version of this manuscript. We have tried to address all concerns raised and we hope the manuscript is now suitable for publication.

REVIEWER COMMENTS

Reviewer #1 (Remarks to the Author):

This excellent paper describes an unusual dominant mutation in the maize WUS1 gene. Although the meristem is fasciated, it is also somewhat barren, unlike the other fasciated ear (fae) mutants in the maize CLV pathway. They identify a tandem duplication that includes WUS and a couple other linked genes. They show that it is the WUS gene by analysis of revertants and generating CRISPR alleles in the dominant mutant allele background. One copy is in the wild-type context (WUS1-A) and the other has a new promoter (WUS1-B).

They use EMS to obtain revertants. One revertant is an EMS mutation in the start codon of WUS1-B. Another revertant has a longer promoter in front of WUS1-B, and 17 others have lost a copy of the duplication. They obtain CRISPR lines that were mutated in both copies. These have normal phenotypes.

They analyze the novel promoter and find it has three copies of a region that is normally present in one copy. This sequence is the binding site for type B ARR proteins. Indeed, they show through EMSAs that ZmRR8 binds this region. They carry out a dual luciferase transient assay in maize protoplasts using a 570 piece of the promoter for WUS1-B. It has stronger activity in the presence of CK than the promoter for WUS1-A (with one copy of the binding region). This is a nice experiment.

They also make transgenics that have a second copy of the gene in its native context. The plants are shorter, but the inflorescence is not different.

The in situs are the most intriguing. They find a hollow sphere through the IM that is devoid of signal. The circle of KN1 and WUS expression is bizarre but the fact it is perpendicular to the axis of the inflorescence is really strange. The serial sections in extended data figure 7 are essential to understand this pattern. I also like the image of *zyb15* at the very tip of the IM. The LOG7 in situs are also stunning. This paper really demonstrates the importance of looking at the pattern of expression with in situs.

They carry out double mutants with CLV mutants. They start this section (line 336) by first describing the SEMS and in situs of *Bif3* in B73. I think they may want to bring these points up in the previous section, or at least start a new paragraph at line 349. It is interesting that *fea3* does not behave the same as *fea2* or *ct2*.

Thank you for the suggestion. We reorganized this section following your advice. It now presents the *Bif3* B73 phenotype before discussing all double mutants. As a result of this, Extended Data Fig. 9 and 10 have been reorganized as well.

Minor points

Line 134. Did they also do qRT-PCR on GRMZM2G047321? Not necessary to do the experiment as they prove it is WUS, but it would be worth noting if they did it.

Yes, we have performed qRT-PCR with GRMZM2G047321, and consistent with the RNA-seq data presented in Fig.2b, the gene is also slightly upregulated (~1.5x). We have not added this data to the manuscript because it is not directly relevant to our results. However, we slightly modified the text as follows:

“Given the enlarged inflorescence meristem phenotype, we concentrated on *ZmWUS1* and confirmed its strong up-regulation by qRT-PCR (Fig. 2c).”

I am curious about the similarity to Ki3. They point to extended data fig 2, but there isn't much there. Did Ki3 also have a duplication or does it have some of this novel promoter?

The similarity to Ki3 was determined based on extensive sequencing of parts of the upstream promoter of the *ZmWUS1A* gene in the mutant. This was done to verify whether potential causative mutations were also present in other inbred lines, before we realized that a tandem duplication existed and before the NAM founder genome sequences became available (this project took a long time ~10 years). Since there were many polymorphisms between the genomic sequence of the *Bif3* mutant and the reference genome B73 we started sequencing the *ZmWUS1* locus in all the NAM founders and that is how we determined that Ki3 was very similar to the *Bif3* background. However, Ki3 does not have a tandem duplication as shown in Extended Data Fig. 2c (at least it does not have the *ZmWUS1B* copy). We have added a new panel with a more extensive set of inbred lines, all lacking *ZmWUS1B* to Extended Data Fig. 2. We also added a new Extended Data Fig. 3 showing the nucleotide alignment of a large fragment (~4.2kb) of the promoter region of the *ZmWUS1A* gene in Ki3, B73 and *Bif3* that shows 99% identity with Ki3 and 60% identity with B73.

We slightly reworded this section as follows:

“The *ZmWUS1-B* gene was only present in the *Bif3* mutant background, and not in a diverse panel of inbred lines (Extended Data Fig. 2). By sequencing several DNA sequence stretches in the region, we determined that Ki3, a Thai inbred line, was very similar, though not identical (~99.1% identity compared to ~60.5% compared to B73; Extended Data Fig.3), to the original *Bif3* mutant background, suggesting that the *Bif3* mutation arose in a closely related tropical-derived line.”

They suggest in the discussion that maize can sustain null mutations in *WUS1* given their CRISPR alleles, however, did they actually analyze plants that were homozygous for these mutations? It seems the analysis was in the context of one normal copy.

We did analyze the phenotype of homozygous *ZmWUS1* knock-out plants and we did not detect any obvious phenotype. One representative plant is shown in Extended Data Fig. 5 (bottom panel) and in this case the plant is heteroallelic for *ZmWUS1* knock-out mutations (1bp deletion and 19bp deletion).

Extended data Fig. 8. It seems the hollow tube is missing in the *Bif3/Bif4* double. That is what caught my attention.

Yes, it does look like the hollow structure is lost in this double mutant as well as in the *Bif3;fea3* double mutant plants we analyzed. The significance of this is unclear at this point.

Line 372. Don't use the word recent twice in one sentence.

Thank you. We fixed it as follows:

“Recent tandem duplications are the most challenging structural variations to identify in genomes and have been significant drivers of phenotypic...”.

The locus reminds me of Kn1-O, which also has a novel promoter and alleles similar to your Rev2. Sarah Hake

Thank you for pointing out this similarity. We added this and another reference in the Discussion:

“However, long chromosome segment repeats are often unstable and lead to repeat losses, as observed for example in other dominant maize mutants^{56,57}.”

Reviewer #2 (Remarks to the Author):

The manuscript "Structural variation at the maize WUSCHEL1 locus alters stem cell organization in inflorescences" reports the cloning and characterisation of a novel WUS mutant in maize for the first time. The authors careful, and meticulous confirmation of the causal mutation shows that genomic rearrangement around the WUS locus caused significant morphological changes in the meristem. This is a well written paper which exhibits a huge quantity of careful investigation of this complex mutant. Their work clearly shows a principal role of WUS gene function in maize meristem regulation in contrast to work in rice, highlighting diversification in meristem regulation across the monocots which could be key in modulation of meristem size to increase yield in different monocot crops.

This is an excellent study and my comments below are for very minor improvements.

- Please include N numbers for the in situ shown

All in situ hybridizations have been repeated at least two times in multiple samples (often, a lot more than that). We have added this information in the Methods section.

“*In situ* hybridization experiments were performed as described⁴⁸ on 2-4mm ear and tassel primordia fixed in PFA, in at least two independent samples with similar results; a representative picture is shown in the figures.”

- Line 118: "As a result, the leaves of Bif3 mutants were 20% wider and 12% shorter than wildtype". I would recommend altering this phrase, as it implies that the authors proved that meristem size has directly altered leaf size.

We agree and modified the text as follows:

“Larger SAMs correlated with leaves of *Bif3* mutants that were 20% wider and 12% shorter than wild type ...”

- Line 132: further clarification on why the gene downstream of WUS, which was also upregulated, was discounted would be beneficial.

We did not discount the downstream gene at first, but incremental evidence led us to concentrate on *ZmWUS1*. Eventually, all evidence pointed to *ZmWUS1* and we confirmed it thanks to the two intragenic suppressor strategies (EMS mutagenesis and CRISPR-Cas9).

We reworded this section as follows:

“Given the enlarged inflorescence meristem phenotype, we concentrated on *ZmWUS1* and confirmed its strong up-regulation by qRT-PCR (Fig. 2c).”

- Extended Data 2: the text refers of panel A showing wild type, Bif3 mutants and revertants, but the panel appear to only be of wild type and Bif3 mutants.

Thank you for catching this mistake. We removed “revertants” since they are not shown in the Figure.

- Line 153: "only present in Bif3 mutant backgrounds and not in diverse inbred lines" refers to Extended Data Figure 2, but it is not clear in the figure that multiple inbred lines have been tested. Perhaps reorder the text?

We have added a new panel in Figure 2 showing the absence of the *ZmWUS1B* locus in different inbred lines.

- Figure 3C: is there an error in R and nR labelling?

No. The ear pictures were taken from plants in which the transgene was segregated away to fix the loss-of-function alleles, while the tassel picture was taken from a T1 plant (generation is specified in Fig. legend).

- In several of the figures a more complete family profile would be valuable in helping highlight the differences between the Bif3 mutants, the wild type of second mutant background and the double mutants, as different stages and tissues are shown.

We have added a few extra panels for the double mutant analysis in which single mutants are shown again. We have also added in Extended Data Fig. 10 pictures of mature single and double mutant tassels with a striking enhancement of the phenotype in particular for *ct2* and *fea3*. We hope we understood and addressed this comment correctly.

- Line 350: the authors state that the ears are enlarged and more fasciated than in single mutants. Is there a way to quantify this difference?

Unlike mild fasciation, extreme fasciation is really difficult to quantify because of the irregular shape of inflorescence meristems. On top of that, it is very hard to compare *Bif3* inflorescence meristems with other meristems, given that in *Bif3* the meristem is deeply embedded in the tissue and axillary meristems often appear on the top of the structure; hence, the visible structure is arguably not a meristem. We have therefore not performed a quantification.

Reviewer #3 (Remarks to the Author):

The manuscript “STRUCTURAL VARIATION AT THE MAIZE WUSCHEL1 LOCUS ALTERS STEM CELL ORGANIZATION IN INFLORESCENCES” presents a beautiful dataset outlining the functional

potency of ZmWus, one of the orthologs of the stem cell regulator WUS.

The manuscript contains a host of genetic, molecular and genomic data to demonstrate that the cause of barren inflorescence3 (Bif3) mutant phenotype lies in the ectopic expression of ZmWUS1. The mechanism for this misexpression is revealed to be a genomic duplication of the ZmWUS locus, along with a multimerization of cytokinin Type-B Response Regulator binding sites. This multiplication of positive regulatory elements is put forward as the cause for WUS transcriptional activation. After having identified the molecular cause of the phenotype they thoroughly characterize the interaction of the mutant with known regulators of maize meristem regulation by genetics and expression studies.

Overall, the manuscript makes a very convincing case to show that ZmWUS1 can affect meristem development and that the phenotype is likely to be driven by cytokinin signalling. The data is of high quality and ample quantity, beautifully presented and well described in the text. Unfortunately, we do not learn much about the *in vivo* function of ZmWUS, nor about its regulation. The authors demonstrate with their CRISPR lines that null alleles of ZmWUS1 have no defect, and the data on cytokinin inducibility shows an exclusive effect on the artificial promoter of the mutant. I do recognize that overexpression studies can make important contributions to our understanding of a biological process that are relevant to a wider audience, but in the complete absence of loss-of-function supported functional evidence this does not seem to be the case here. The authors argue that functional redundancy with ZmWUS2 is masking the effect of removing ZmWUS1, but the diverging expression domains of both genes would argue otherwise. Since the authors already have generated null alleles for ZmWUS1, I strongly recommend to use the same strategy to produce ZmWUS2 null mutants for double mutant analysis. I am afraid that without a clear loss-of-function phenotype for ZmWUS, the manuscript will only appeal to a rather small group of WUS enthusiasts and hence will remain suitable only for much more specialized journals.

The reviewer is correct in pointing out that the expression of *ZmWUS1* and *ZmWUS2* were reported to differ in Nardmann and Werr, 2006 (Nardmann and Werr, *Mol Biol and Evol*, 23:2492-2504, 2006) based on *in situ* hybridizations. However, the same paper shows clear overlapping signals of both genes (in Figure 5) in all axillary meristems, although not in the inflorescence meristem. Recent data from a ZmWUS1 transcriptional reporter construct (Je et al *Nat Gen*, 48:785-791, 2016) show that *ZmWUS1* is expressed in the inflorescence meristem (data that agree with our *in situ*), albeit at levels that are low and therefore may have escaped earlier detection. We believe that this discrepancy is due to the limit of *in situ* hybridizations, as in our hands it is very difficult to get a decent signal of *ZmWUS1* in *in situ* hybridizations of inflorescence meristems in normal plants. Hence, if we reconcile all the data known so far, it is quite reasonable to suspect redundancy in the function of *ZmWUS1* and *ZmWUS2*. This is actually something we are very interested in pursuing and have recently received funding to investigate, including revisiting the expression differences between the two genes. However, this will take at least two years (including introgressions in appropriate genetic backgrounds), and we believe it is outside the scope of this manuscript.

We added a sentence in the discussion:

“In support of its role in meristem size regulation, our data show that *ZmWUS1* is expressed at a very low level in IMs and agree with data from a published transcriptional reporter line ²³”

We disagree that our findings only appeal to WUS enthusiasts. We believe this is a beautiful example of how multimerization of a single binding site recognized by a specific transcription factor family (in this case type-B RRs) can have a major effect on development. There are very few such instances described in the literature. Also, data on cytokinin inducibility do not show an exclusive effect on the mutant promoter; the effect is seen on wild type promoter as well, just to a minor extent (see Fig. 3j). This highlights how adding more copies of the same TFBS increases transcriptional output and raises intriguing questions about how type-B RRs promote transcription i.e. are multimerized sites bound by multimers of type-B RRs? Is spacing among binding sites important?

Minor comment: The illustration in Fig4 e does not correlate with the WUS expression domain shown in panel a. The data seem to suggest that WUS mRNA can be found at the apical dome, but the illustration suggests otherwise. Please also discuss published data on ZmWUS expression, they do seem to differ.

This is a good point, thank you for pointing out this discrepancy. We agree that the expression pattern of *ZmWUS1* in inflorescence meristems (based on our *in situ* data and on the reporter line mentioned above) does not appear to be as deep in the tissue as in Arabidopsis meristems. We revised the model accordingly in a new version of Fig. 4e.